# Phase-Field Insights into Hydrogen Trapping by Secondary Phases in Alloys

**DOI:** 10.3390/ma16083189

**Published:** 2023-04-18

**Authors:** Shijie Bai, Lin Liu, Chenyang Liu, Chao Xie

**Affiliations:** The Faculty of Mechanical Engineering and Mechanics, Ningbo University, Ningbo 315211, China; 2011081001@nbu.edu.cn (S.B.); 2111081149@nbu.edu.cn (L.L.); 2111081014@nbu.edu.cn (C.L.)

**Keywords:** hydrogen trapping, hydrogen storage, secondary phase, alloy, phase-field modeling

## Abstract

Solid-state hydrogen storage is the best choice for balancing economy and safety among various hydrogen storage technologies, and hydrogen storage in the secondary phase might be a promising solid-state hydrogen storage scheme. In the current study, to unmask its physical mechanisms and details, a thermodynamically consistent phase-field framework is built for the first time to model hydrogen trapping, enrichment, and storage in the secondary phases of alloys. The hydrogen trapping processes, together with hydrogen charging, are numerically simulated using the implicit iterative algorithm of the self-defined finite elements. Some important results are attained: 1. Hydrogen can overcome the energy barrier under the assistance of the local elastic driving force and then spontaneously enter the trap site from the lattice site. The high binding energy makes it difficult for the trapped hydrogens to escape. 2. The secondary phase geometry stress concentration significantly induces the hydrogen to overcome the energy barrier. 3. The manipulation of the geometry, volume fraction, dimension, and type of the secondary phases is capable of dictating the tradeoff between the hydrogen storage capacity and the hydrogen charging rate. The new hydrogen storage scheme, together with the material design ideology, promises a viable path toward the optimization of critical hydrogen storage and transport for the hydrogen economy.

## 1. Introduction

With the increasing demand for carbon neutrality and sustainable development [1,2,3], hydrogen, as a clean energy source, has received considerable attention. In recent years, hydrogen energy is regarded as the most promising secondary energy source [4,5] owing to its higher energy density, abundance in quantity, pollution-free and zero-emission characteristics, storability, renewability, and so on. However, the major obstacle that hinders the development of the hydrogen economy is storage and transport [6]. If a suitable solution for storing hydrogen is found, hydrogen will ascend to be an alternative to fossil fuel. As shown in Figure 1, various hydrogen storage technologies have been progressively developed. Nowadays, mainstream technologies can fall into three categories: high-pressure gaseous hydrogen storage, liquid hydrogen storage, and solid-state hydrogen storage [7]. Among them, high-pressure gaseous hydrogen storage has low bulk hydrogen density and poor safety performance. Liquid hydrogen storage has high costs, high loss, and high requirements for hydrogen storage containers. So far, solid-state hydrogen storage seems to be the best choice under the comprehensive consideration of economy and safety. Both the low hydrogen storage density per unit mass and low efficiency of hydrogen charging and discharging limit the popularization of this technology. In addition, a flexible media for the storage and transportation of hydrogen, the liquid organic hydrogen carriers (LOHCs), provides a commercially feasible and mature concept [8,9]. These “liquid hydrogen batteries” use catalysts at high temperatures to realize reversible hydrogenation and dehydrogenation.

A massive effort has been devoted to the development of solid-state hydrogen storage. Besides the active investigations on hydrides [10,11], the latest study [12] indicates that the secondary phases within aluminum alloys are capable of trapping large amounts of hydrogens (Figure 2c), which implies a potential new type of hydrogen storage scheme. Hydrogens enter the metal and are neatly aligned on the metal lattice; however, the hydrogens in the vicinity of the secondary phases might overcome the energy barrier (Figure 2b) and be bound at the trapping sites (Figure 2a) owing to the eigenstrain of the lattice coherency and the dilatation caused by hydrogen entering the lattice or other external effects. The trapping sites, acting as sinks, attract the lattice hydrogens, leading to the enrichment of hydrogen in the secondary phases.

To quantitatively investigate trapping mechanisms, some scholars have modeled and simulated the transport and trapping of hydrogen in metals. Alfons et al. [13] derived a hydrogen trapping model which resembles McNabb [14] and Oriani’s models [15] but was not described as a thermodynamic process. Zhou et al. [16] used the FE (finite element) method to simulate hydrogen transport and hydrogen segregation in the vicinity of secondary phases under stress, and the effects of microcracks, stress magnitudes, as well as the shape and orientation of secondary phases on hydrogen segregation were investigated. However, the physical process of hydrogen trapping was not really included; Fernández-Sousa et al. [17] developed a mechanistic, multi-trap model for hydrogen transport, implemented it into an FE framework, and captured the variations in crack tip lattice and trapped hydrogen concentrations, but the model did not involve the intrinsic mechanism of hydrogen entering trapping sites. The abovementioned modeling, which is not completely compatible with the thermodynamic mechanism of the hydrogen trapping in secondary phases, could not really capture the physical process of the hydrogen trapping and might miss some details that can motivate the design of the new hydrogen storage scheme. In addition, Leitner et al. [18] physically modeled the trapping and diffusion of multiple solute atoms in systems with multiple traps based on irreversible thermodynamics and the representative volume element technique (RVE), covering effectively any kinetics of exchange between the lattice and traps as well as site competition effects within traps for any system size. Unfortunately, the trapping effect of secondary phases on hydrogens was not included.

In hydrogen transport, the phase-field method has been widely used due to the diffusion interface which avoids the difficulty of tracking the interface at conventional sharp interfaces. Martínez-Pañeda et al. [19] presented a phase-field modeling framework for hydrogen-assisted cracking, which builds upon a coupled mechanical and hydrogen diffusion response driven by chemical potential gradients and a hydrogen-dependent fracture energy degradation law grounded on first-principle calculations. Kristensen et al. [20] developed a gradient-based theoretical phase-field framework for predicting hydrogen-assisted fracture in elastic-plastic solids. Wu et al. [21] proposed a phase-field regularized cohesive zone model for hydrogen-assisted cracking. However, none of these involves the hydrogen trapping process.

The purposes of this study are to physically model the hydrogen trapping in the secondary phases adopting the phase-field paradigm based on the continuum mechanics and irreversible thermodynamics and to provide insights into the material design of the new hydrogen storage scheme.

## 2. Materials and Methods

In this study, the mechanical equilibrium (Momentum conservation)-hydrogen trapping (Nonconservative phase field)-lattice hydrogen transport (mass conservation) coupled framework is established. The mechanical, chemical, hydrogen-trapping, and interfacial components of the Helmholtz free energy are first proposed. According to the previous work [22], the weak forms of the governing equations for the coupled field can then be obtained. Finally, the dynamic evolution process of hydrogen trapping can be numerically solved using the implicit integration scheme of FE methods with Newton–Raphson iteration [22].

Some important details for the modeling need to be clarified:(1)The material is assumed to be isotropic linear elastic, and the external mechanical load is not considered in the model.(2)To clearly present the effect of lattice mismatch, the mismatch of elastic property, diffusive coefficient, and solubility between the matrix and secondary phase is ignored.(3)The hydrogen trapping is treated as a thermodynamic process using the phase-field method. The diffusive interface is employed to describe the moving interface between trapping and non-trapping statuses.

### 2.1. Multi-Physical Coupled Field

The current study aims to investigate the thermodynamically irreversible process of hydrogen trapping under the local elastic energy. The conservation of momentum, phase-field description for the nonconservative trapping process, and conservation of mass can be formulated as follows:

Generalized equilibrium equations:(1)$$\begin{array}{l} \nabla \cdot \sigma =0\;\;\;\;\;\;\;\;\;\;\;\;\;\left(a\right)\\ \left(\nabla \cdot \zeta -\omega \right)-\frac{1}{L}\frac{d\varphi }{dt}=0\;\;\;\;\;\;\;\;\left(b\right)\\ \frac{dc_{L}}{dt}c_{\mathrm{sat}}+\nabla \cdot J-Q=0\;\;\;\;\;\;\;\;\;\;\;\left(c\right) \end{array}$$

Generalized force boundary conditions:(2)$$\begin{array}{l} T=\sigma \cdot n\;\;\;\;\;\;\;\;\;\;\left(a\right)\\ f=\zeta \cdot n\;\;\;\;\;\;\;\;\left(b\right)\\ q=J\cdot n\;\;\;\;\;\;\;\;\left(c\right) \end{array}$$
where σ is the stress tensor, T is the surface force, and n is the surface normal unit vector; L is the interfacial kinetic coefficient, the negative driving force ω and the negative flux ζ conjugate to the phase field φ and phase-field gradient ∇φ, respectively; φ is the occupancy of the trapping sites, φ=0 means that no hydrogen is trapped for the material point, φ= 1 means that all trapping sites for the material point are fully occupied, and f is the phase-field micro-traction; and $c_{L}$ is the dimensionless concentration of lattice hydrogen in crystal, $c_{\mathrm{sat}}=\frac{N_{L}}{N_{A}}$ is the saturation concentration of lattice hydrogen in Al metal, $N_{L}$ is the number of lattice sites per unit volume, $N_{A}$ is the Avogadro constant, J is the mass flux, q is the incoming concentration flux, and $Q=-\dot{\varphi }\frac{N_{T}}{N_{A}}$ is defined as the sink of the lattice hydrogen concentration field.

In addition, the total hydrogen concentration is defined as $c_{\mathrm{total}}=c_{L}c_{\mathrm{sat}}+c_{T}$, where $c_{T}=\varphi \frac{N_{T}}{N_{A}}$ is the trapped hydrogen concentration in which $N_{T}$ is the number of trapping sites per unit volume.

The details of the coupled system are depicted in Figure 3: (1) the local elastic energy resulting from the hydrogen-trapping dilatation assists the hydrogen to overcome the energy barrier and enter the trapping sites ($\omega \left(\sigma \right)$); (2) the occupancy of trapping and lattice sites results in the variation in stress ($\sigma \left(c_{L},\varphi \right)$); and (3) the lattice hydrogen transport is dominated by the concentration gradient and the hydrostatic pressure gradient $J\left(\nabla c_{L},\;\nabla \sigma_{m}\right)$ and the sink caused by hydrogen trapping ($Q\left(\varphi \right)$).

### 2.2. Helmholtz Free Energy

The total Helmholtz free energy density for the system can be decomposed into the lattice elastic, hydrogen-trapping, chemical, and interfacial components.

The lattice elastic energy density $\Phi^{e}$ is defined as:(3)$$\Phi^{e}=\frac{1}{2}\left(\varepsilon -\varepsilon_{\mathrm{lm}}\right):C:\left(\varepsilon -\varepsilon_{\mathrm{lm}}\right)$$
where ε represents the total strain tensor and $\varepsilon_{\mathrm{lm}}$ represents the eigenstrain tensor.

For the hydrogen trapping system, the trapping free energy $\Phi^{L\to T}$ of hydrogen entering the trapping site from the lattice site is proposed for the first time as:(4)$$\Phi^{L\to T}=E_{b}\left(1+\alpha \frac{\varphi^{2}}{2}-\beta \frac{\varphi^{3}}{3}+\gamma \frac{\varphi^{4}}{4}\right)$$
where $E_{b}$ is the binding energy, β=3α+12, γ=2α+12, and $\alpha =32\frac{E_{s}}{E_{b}}$, in which $E_{s}$ is the saddle point energy.

The chemical free energy density $\Phi^{\mathrm{ch}}$ can be further decomposed into the component associated with the concentration and that associated with the local elastic dilatation:(5)$$\Phi^{\mathrm{ch}}=\mu_{0}c_{\mathrm{sat}}c_{L}+2A\left(-c_{L}+c_{L}\ln c_{L}\right)-KV_{\mathrm{hl}}c_{\mathrm{sat}}c_{L}\mathrm{tr}\left(\varepsilon -\varepsilon_{\mathrm{lm}}\right)-KV_{\mathrm{ht}}\frac{N_{T}}{N_{A}}\varphi \mathrm{tr}\left(\varepsilon -\varepsilon_{\mathrm{lm}}\right)$$
where A is the free energy density curvature, K is the bulk modulus, and $V_{\mathrm{hl}}$ and $V_{\mathrm{ht}}$ are the partial molar volume for the lattice hydrogen and the trapping hydrogen, respectively.

In addition, the interfacial free energy density $\Phi^{i}$ is defined as a function of the gradient of the phase-field order parameter as follows:(6)$$\Phi^{i}=\frac{k_{g}G_{c}l_{c}}{2}{\left|\nabla \varphi \right|}^{2}$$
where $G_{c}$ is the surface energy density, $l_{c}$ is the interfacial characteristic thickness, and $k_{g}$ is a normalized constant that controls the width of the diffusive interface.

Substituting the given free energy densities into the constitutive relationships [22], the governing equations for the coupled field can be obtained.

### 2.3. FE Implementation

The user element subroutine (UEL) of the commercial FE software ABAQUS 6.14 is employed to define the discretized elements for the coupled field of hydrogen trapping.

The weak forms of the governing equations, residuals, and simplified tangent stiffness matrices can be obtained according to the previous work [22]. It is worth noting that the gradient of the elastic hydrostatic pressure is given as:(7)$$\nabla \sigma_{m}=K\nabla \mathrm{tr}\left(\varepsilon -\varepsilon_{\mathrm{lm}}\right)=KB_{}^{s}\hat{u}\;$$
where K is the bulk modulus, $B^{s}{}_{\left(2\times 2i\right)}$ is the second gradient matrix, and ${\hat{u}}_{\left(2i\times 1\right)}$ is the nodal displacement matrix.

## 3. Results

### 3.1. Justification of Free Energy

By nature, thermodynamic systems, including the hydrogen trapping system, tend to the minimum energy state. As shown in Figure 4, the lattice hydrogen at the stable equilibrium status (φ=0) can overcome the energy barrier $E_{s}$ under a driving force, and then spontaneously converge to the trapping site (φ= 1) due to the principle of minimum energy. It is relatively more difficult for the hydrogen to escape from the trapping site and return to the lattice site owing to the much higher binding energy. The factors that can decrease the binding energy and/or provide enough external driving force for the hydrogen to overcome the high energy barrier are required. This concerns how hydrogens are released after storage. In addition, it is worth noting that the driving force is composed of the local elastic energy resulting from the hydrogen-trapping dilatation and the negative gradient of trapping free energy. Before the saddle point energy (energy barrier) is overcome, the elastic effect and free energy effect provide positive and negative contributions to the trapping, respectively. As shown in Figure 5, the excessive $E_{s}$ is inclined to lead to the negative driving force, preventing the hydrogen from overcoming the energy barrier and entering the trapping site at the same stress level.

### 3.2. Hydrogen Trapping without Local Elastic-Driven Force

In the current study, all models, with the material parameters given in Table 1, are discretized using quadrilateral quadratic plane strain elements.

During metal processing and service, chemical reactions that produce small amounts of hydrogen may occur. With the assistance of external effects, the hydrogen at lattice sites may overcome the energy barrier. Based on the principle of minimum energy, the hydrogen that overcomes the energy barrier tends to a lower energy state, hence entering the trapping site and forming the pre-trapped hydrogen point.

The simulation of a rectangular Al single crystal containing a circular secondary phase with pre-trapped points was conducted to investigate the evolution of hydrogen trapping. The model dimensions and initial conditions are presented in Figure 6a, and the FE mesh is shown in Figure 6b.

The materials used in this study are 7075 Al alloys. The proposed approach is not specific to hydrogen trapping by a particular secondary phase. The effect of the variations in $E_{b}$ and $E_{s}$ on hydrogen trapping were previously discussed in Section 3.1. Because some parameters are absent and hard to determine, the values from Fernández-Sousa et al. [15] are employed in this study to investigate the regulation and mechanism of hydrogen trapping.

As shown in Figure 7a, under the driving force caused by the trapping free energy, the spare trapping sites at the pre-trapped point (φ= 0.6) can be quickly occupied because lots of lattice hydrogens at the point continuously enter the trapping sites, and thus the lattice hydrogens in the adjacent area diffuse toward to the point. Meanwhile, in the adjacent area, lattice hydrogens are also inclined to overcome the energy barrier and enter the trapping sites. Therefore, the hydrogen trapping activity rapidly spreads outward and finally occupies the whole secondary phase. As shown in Figure 7b, the concentration of lattice hydrogen is clearly reduced owing to the prevailing hydrogen trapping in the secondary phase. A massive amount of lattice hydrogens in the matrix continuously diffuse toward the secondary phase until the new concentration equilibrium is reached. Figure 7c shows that the total hydrogen concentration of the secondary phase is a dozen times higher than that of the matrix. The hydrogen segregation caused by the prevailing trapping results in clear hydrogen enrichment.

### 3.3. Hydrogen Trapping with Local Elastic-Driven Force

Hydrogen trapping is prone to be triggered by four types of driven forces, such as the attraction of electron vacancy in metal, lattice elastic energy gradient induced by crystal defects, thermal gradient, and chemical potential gradient [23]. It is worth noting that the trapping driven by the local elastic energy induced by the lattice stress resulting from crystal mismatch/defects, such as lattice interfacial coherence, dislocations, grain boundaries, secondary phases, crack tips, and so forth, is the most common. In the current study, the local elastic energy resulting from the hydrogen-trapping dilatation is considered to assist hydrogen trapping from the trapping-free status.

To investigate the effect of lattice stress on the hydrogen trapping process, a rectangular Al single crystal containing a secondary phase with the coherent strain was simulated. The model dimensions, initial conditions, and Dirichlet-type fixed boundary condition are presented in Figure 8a, and the FE mesh is shown in Figure 8b.

As shown in Figure 9, the tensile stress field caused by the coherent eigenstrain appears in the secondary phase and near the interface. The local elastic energy resulting from the hydrogen-trapping dilatation is capable of assisting lattice hydrogens to overcome the energy barrier and eventually realizes the hydrogen trapping from the trapping-free status. The earliest position of hydrogen trapping is at the interface between the secondary phase and the matrix (Figure 10b), where there is the greatest stress concentration (Figure 9a and Figure 10a). Then, the lattice hydrogens at the adjacent points on the interface and within the secondary phase also overcome the energy barrier and enter the trapping sites. In addition, the lattice hydrogens in the matrix are attracted by the sink of lattice hydrogens in the secondary phase, which achieves a new concentration equilibrium (Figure 10c). Finally, substantial hydrogens are trapped in the secondary phase (Figure 10d), which leads to the apparent hydrogen segregation and enrichment within the secondary phase. Recently, Zhao et al. [12] performed a near-atomic-scale analysis of the hydrogens trapped in secondary phases in a high-strength 7xxx Al alloy and found that a dozen times more hydrogens were segregated in the secondary phases than those in the matrix. Therefore, the obtained numerical result in the current study is in good agreement with the experiment of Zhao et al. in Figure 2c.

Furthermore, to demonstrate the influence of the local elastic energy resulting from the hydrogen-trapping dilatation on the hydrogen-trapping process, the hydrogen trapping in the secondary phase with the same area and different geometries are simulated. The model dimensions, initial conditions, and Dirichlet-type fixed boundary condition are presented in Figure 11. Three secondary phases have different ratios of the short axis to the long axis: 1:1, 4:9, and 1:9, respectively.

As shown in Figure 12a, the smallest ratio of the short axis to long axis leads to the most intensive stress concentration at the ends of the secondary phase with the greatest curvature. Thus, the hydrogen trapping appears at the interfacial sites the earliest and spreads rapidly all over the whole secondary phase (Figure 12d). The greater the stress concentration is, the earlier the hydrogens overcome the energy barrier and are trapped in the secondary phase (Figure 12b–d). Therefore, the geometry of the secondary phase is closely related to the hydrogen trapping. For alloys, reasonable defect engineering, especially precipitation morphology design, is promising to improve the hydrogen charging rate.

### 3.4. Hydrogen Charging

The continuous rise of the hydrogen energy economy significantly relies on credible storage technologies [24,25,26,27]. For solid-state hydrogen storage, the major problems are the low storage density and low efficiency of hydrogen charging and discharging. Therefore, for the proposed new hydrogen storage scheme, the investigation of the effect of secondary phases on the rate of hydrogen charging is of primary importance. The effects of volume fraction and refinement of the secondary phases on hydrogen charging should be analyzed.

#### 3.4.1. Volume Fraction

To investigate the relationship between the volume fraction of secondary phases and the hydrogen storage capacity, the hydrogen charging process of Al crystals with 1, 2, 4, and 8 secondary phases is simulated. The model dimensions, initial conditions, and Dirichlet-type fixed boundary condition are presented in Figure 13. In addition, the Dirichlet boundary conditions $c_{L}=\;1$ are imposed on the lower surface to model the hydrogen-charging condition.

As shown in Figure 14, the trapped hydrogen content increases with increasing the volume fraction of the secondary phases. After the beginning of hydrogen charging, the charged hydrogen first spreads within the matrix and moves along the inverse direction of the lattice hydrogen concentration gradient. Subsequently, the hydrogen enters the secondary phase, and the local elastic energy assists the hydrogen in the secondary phase to overcome the energy barrier and be trapped at the trapping site. The trapping process acts as a sink attracting a large number of lattice hydrogens in the vicinity of the secondary phase to enter and be trapped in the secondary phase. Finally, the saturated states of the matrix and secondary phase are attained. It should be noted that the increase in the secondary phase volume fraction has no effect on the saturated concentration of the hydrogen storage but significantly enhances the solid-state hydrogen storage capacity of the crystal.

#### 3.4.2. Refinement

The model dimensions, initial conditions, and Dirichlet-type fixed boundary condition are presented in Figure 15. In addition, the Dirichlet boundary conditions are imposed on the lower surface to model the continuous hydrogen charging condition.

To characterize the effect of secondary phases on the hydrogen charging rate, the equivalent hydrogen diffusive coefficient D is estimated using the second Fick’s law solution [28].
(8)$$\frac{c(x)-c_{s}}{c_{0}-c_{s}}=erf\left(\frac{x}{2\sqrt{2.4Dt}}\right)$$
where c(x)=0.9 is the hydrogen concentration at a distance x from the hydrogen charging surface, t is the time, $c_{s}=1$ is the lattice hydrogen concentration at the charging surface, and $c_{0}=\;0$ is the initial lattice hydrogen concentration of the model. The initial diffusive coefficient of hydrogen of the Al alloy without the secondary phases is $D_{0}=\;2.2\times 10^{-8}{\;\mathrm{mm}}^{2}/s$. The dimensionless equivalent diffusive coefficient is defined as $D_{\mathrm{eq}}=\frac{D}{D_{0}}$.

As shown in Figure 16, during the hydrogen charging process, hydrogen gradually diffuses upwards from the bottom of the crystal. At the comparatively sharper ends of the ellipsoidal secondary phases, greater local elastic energy is induced due to the greater stress concentration (Figure 16a). Driven by both the local stress energy and negative gradient of trapping free energy, the lattice hydrogens at the ends first overcome the energy barrier and enter the trapping site. The mature trapping facilitates more hydrogen entering the trapping sites at the interface and in the secondary phase (Figure 16b). Moreover, since the hydrogen charging takes place at the bottom of the crystal, the lower secondary phases trap hydrogens faster than the upper ones. Therefore, under the same eigenstrain conditions, the trap sites of the lower secondary phases are filled with priority, making the lattice hydrogen concentration non-uniform during hydrogen charging (Figure 16c). However, hydrogen enrichment in the refined secondary phases is achieved after the hydrogen charging complement (Figure 16d).

Figure 17 shows that the appearance of the single secondary phase significantly reduces the equivalent diffusive coefficient, delaying the hydrogen charging process. The increasing amount of trapping sites absorbs more hydrogens during the hydrogen charging process, leading to the lower equivalent diffusive coefficient. In addition, the equivalent diffusive coefficient is affected by the refinement of secondary phases to some extent.

Therefore, the manipulation of secondary phase geometry, volume fraction, and dimensions is critical to the tradeoff between the storage capacity and charging rate.

## 4. Discussions

### 4.1. Effect of Secondary Phase Type on Hydrogen Charging Rate

For alloys, there are a variety of secondary phases, which have different numbers of trapping sites per unit volume $N_{t}$. The relationships between the equivalent diffusive coefficient $D_{\mathrm{eq}}$ and the number of trapping sites per unit volume $N_{t}$ needs to be clarified. The model with the initial and boundary conditions of Figure 15b is employed.

The results show that (Figure 18) the $D_{\mathrm{eq}}$ first dramatically drops with increasing ratio $N_{t}/N_{t0}$, wherein $N_{t0}=7\times 10^{19}{\;m}^{-3}$, and then decreases at a relatively low speed. This means that the presence of secondary phases can significantly reduce the hydrogen charging rate, and the subsequent increase in trapping sites gradually slows down the hydrogen charging rate.

### 4.2. Negligible Effect of Hydrostatic Pressure Gradient

In most studies regarding hydrogen transport, the hydrostatic pressure gradient is generally considered in the model with cracks. In the current study, it is numerically found that the effect of the hydrostatic pressure gradient on the lattice hydrogen transport is very weak (Figure 19). The values of the gradient components are around five orders of magnitude less than that in the model with cracks. Therefore, the effect of hydrostatic pressure gradient on the hydrogen transport is ignored in this study.

## 5. Conclusions

A phase-field paradigm based on the continuum mechanics and irreversible thermodynamics is developed for the trapping, enrichment, and storage of hydrogen in the secondary phases of alloys. According to the ABAQUS UEL subroutine that describes the coupled constitutive relationship and coupled element, the hydrogen trapping with/without the local elastic driven force and the hydrogen charging of the Al crystal with different secondary phases are numerically simulated. The justification of the free energy reveals the physical mechanism of hydrogen trapping. The simulation results provide some mechanistic insights into the hydrogen storage scheme. Some conclusions can be drawn:(1)For the secondary phase with pre-trapped points, the hydrogen segregation caused by the pre-trapped point results in clear hydrogen enrichment. For the secondary phase with the eigenstrain, the local elastic energy resulting from the hydrogen-trapping dilatation is generated and assists the hydrogen trapping from the trapping-free status. The high binding energy significantly hinders the hydrogen escape from the trap sites. However, the binding energy might be susceptible to temperatures and other factors, which offers the possibility of hydrogen release for the new hydrogen storage scheme.(2)Keeping the secondary phase volume fraction constant and varying the ratio of the short axis to the long axis, it is found that the ends with greater curvatures are prone to cause stronger stress concentrations which induce the greater local elastic energy to drive the hydrogen, thus overcoming the energy barrier.(3)For hydrogen charging, the increase in the volume fraction of the secondary phases can enhance the hydrogen storage capacity effectively. The refinement of secondary phases can reduce the hydrogen charging efficiency. The increase in trapping sites also gradually slows down the hydrogen charging rate. The volume fraction, refinement, and type of secondary phases can be quantitatively regulated by the alloying and heat treatment processes, enabling the optimization of the hydrogen charging.

## Figures and Tables

**Figure 1 materials-16-03189-f001:**
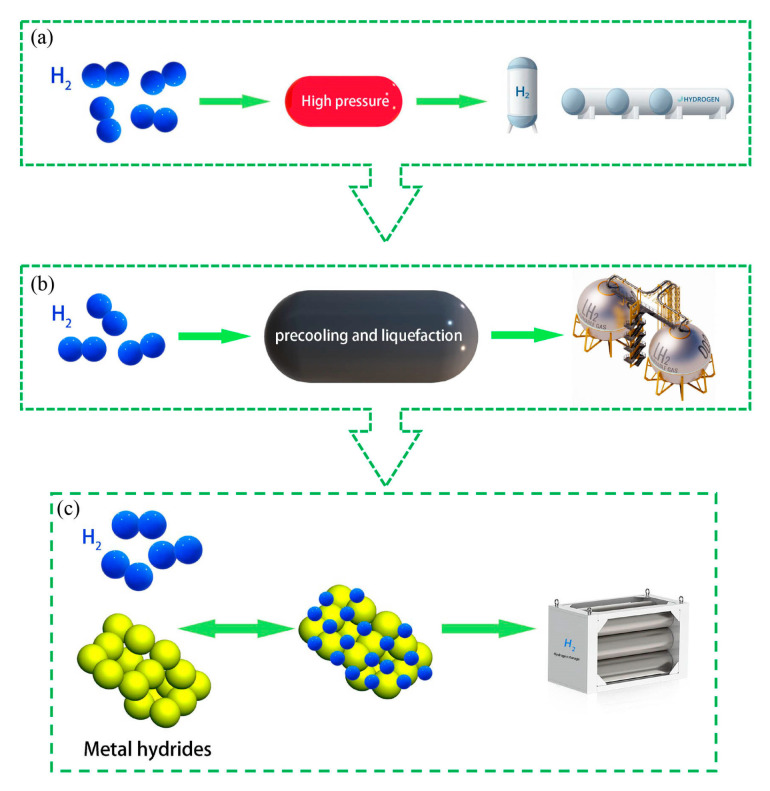
Schematic of hydrogen storage technologies: (**a**) high-pressure gaseous hydrogen storage; (**b**) liquid hydrogen storage; and (**c**) solid-state hydrogen storage.

**Figure 2 materials-16-03189-f002:**
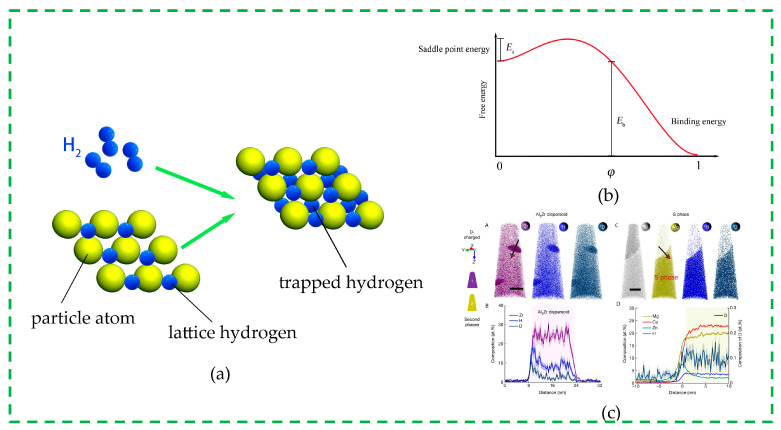
Hydrogen trapping in secondary phases within aluminum alloys: (**a**) schematic diagram of the new hydrogen storage method; (**b**) free energy curve for hydrogen trapping; and (**c**) experimental measurement of hydrogen segregation [12], Atom map and composition profiles are presented along the red arrows respectively for Al_3_Zr dispersoids (A, B) and S phase (C, D).

**Figure 3 materials-16-03189-f003:**
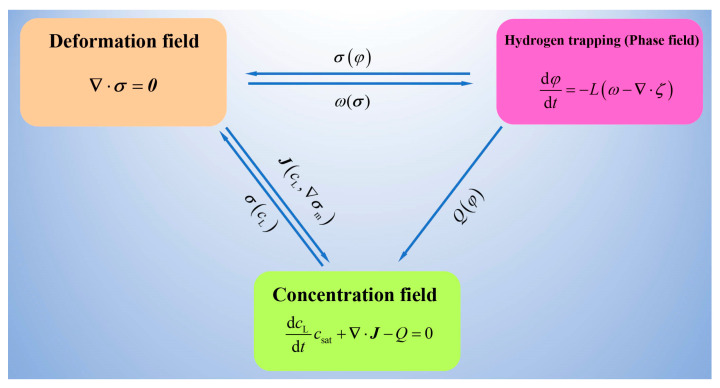
Schematic of multi-physical field coupling.

**Figure 4 materials-16-03189-f004:**
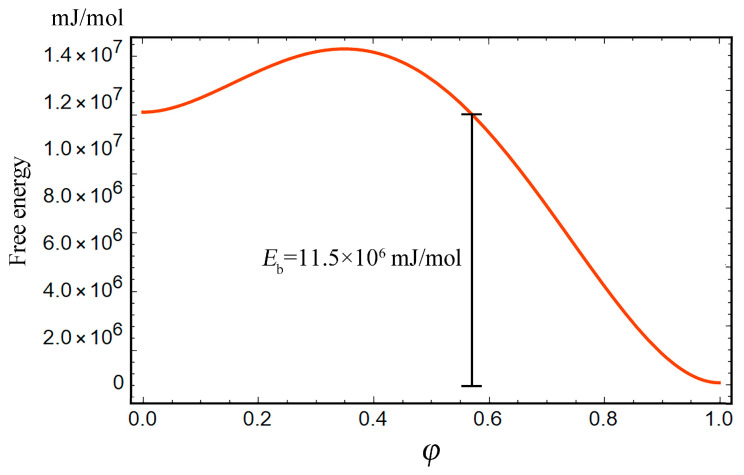
Free energy curve of hydrogen trapping system.

**Figure 5 materials-16-03189-f005:**
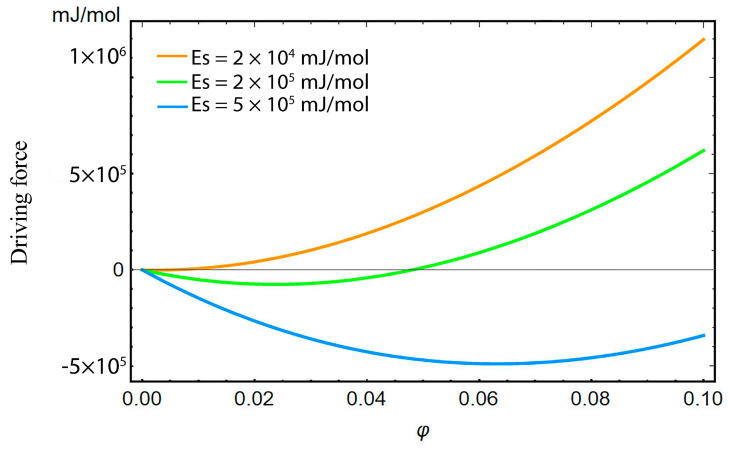
Driving force of hydrogen trapping system with different saddle point energy.

**Figure 6 materials-16-03189-f006:**
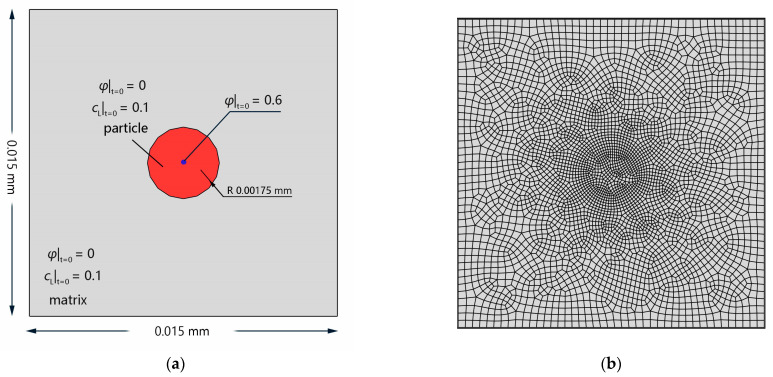
FE model without eigenstrain: (**a**) schematic description of initial and boundary conditions of an Al single crystal with one trapped point in the secondary phase; (**b**) FE mesh.

**Figure 7 materials-16-03189-f007:**
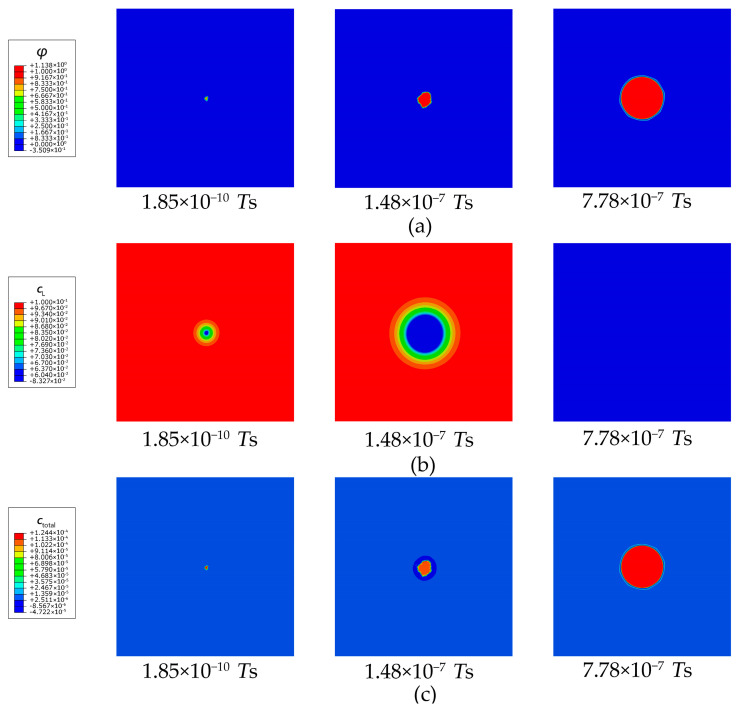
Evolution of (**a**) occupancy of hydrogen trapping site φ, (**b**) lattice hydrogen concentration $c_{L}$, and (**c**) total hydrogen concentration $c_{\mathrm{total}}$ for a pre-trapped point in the secondary phase of an Al single crystal.

**Figure 8 materials-16-03189-f008:**
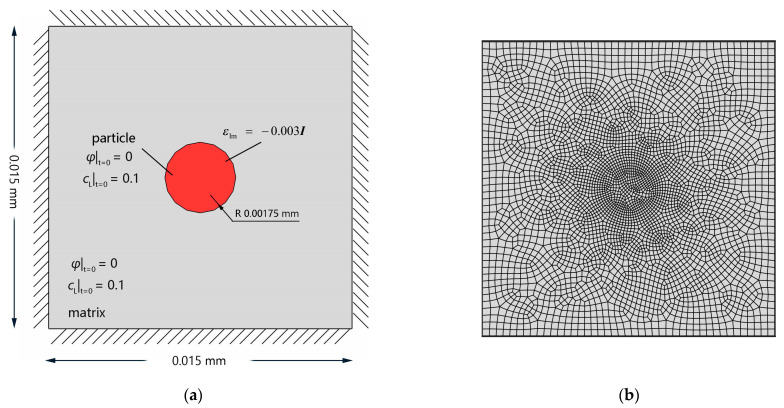
FE model with eigenstrain: (**a**) schematic description of initial and boundary conditions of an Al single crystal with a single secondary phase; (**b**) FE mesh.

**Figure 9 materials-16-03189-f009:**
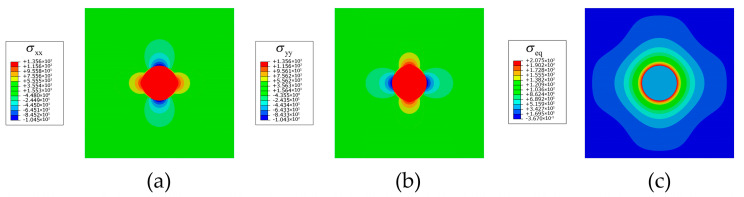
Stress field (unit: MPa) in Al with secondary phase under eigenstrain: (**a**) $\sigma_{\mathrm{xx}}$, (**b**) $\sigma_{\mathrm{yy}}$, and (**c**) $\sigma_{\mathrm{eq}}$.

**Figure 10 materials-16-03189-f010:**
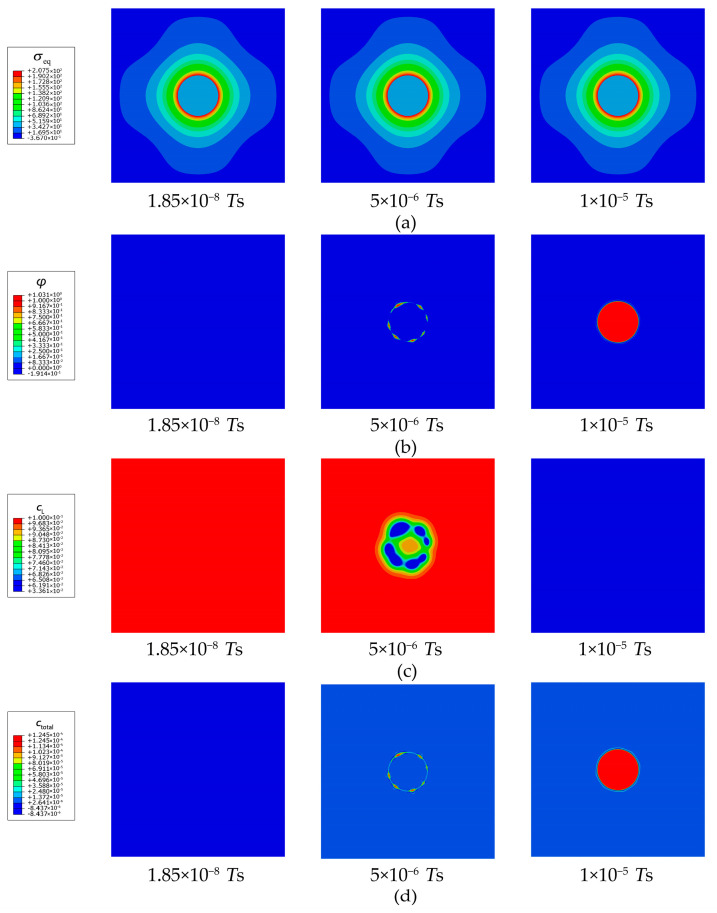
Evolution of (**a**) equivalent stress (unit: MPa), (**b**) occupancy of hydrogen trapping site φ, (**c**) lattice hydrogen concentration $c_{L}$, and (**d**) total hydrogen concentration $c_{\mathrm{total}}$ for an Al crystal with a single secondary phase.

**Figure 11 materials-16-03189-f011:**

Schematic description of initial and boundary conditions of an Al single crystal with (**a**): a circular secondary phase; (**b**) an elliptical secondary phase with the ratio of the short axis to the long axis 4:9; and (**c**) an elliptical secondary phase with the ratio of the short axis to the long axis 1:9.

**Figure 12 materials-16-03189-f012:**
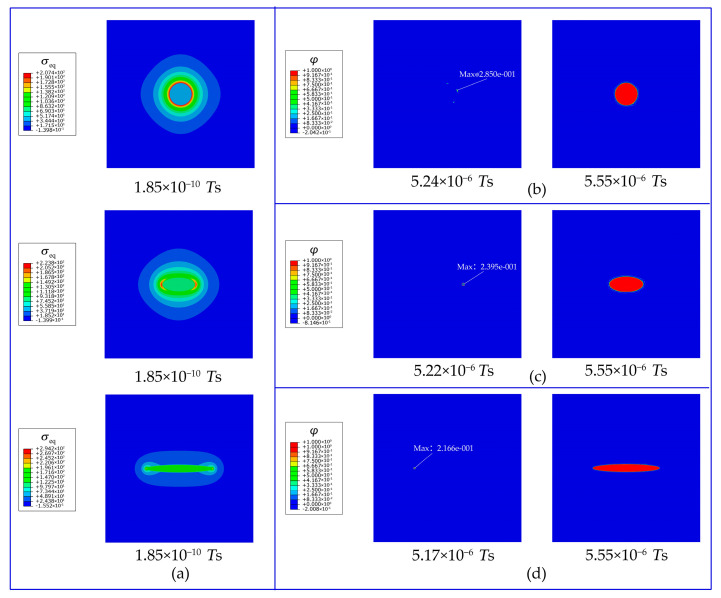
Evolution of (**a**) equivalent stress (unit: MPa), (**b**) occupancy of hydrogen trapping site φ, (**c**) lattice hydrogen concentration $c_{L}$, and (**d**) total hydrogen concentration $c_{\mathrm{total}}$ for an Al crystal with a single secondary phase.

**Figure 13 materials-16-03189-f013:**
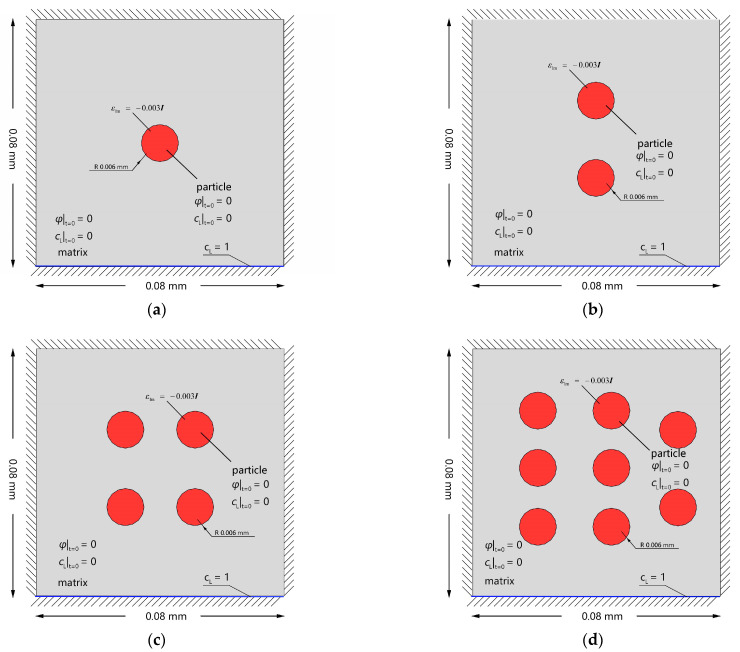
Schematic description of initial and boundary conditions of Al single crystals with (**a**) 1 secondary phase; (**b**) 2 secondary phases; (**c**) 4 secondary phases; and (**d**) 8 secondary phases.

**Figure 14 materials-16-03189-f014:**
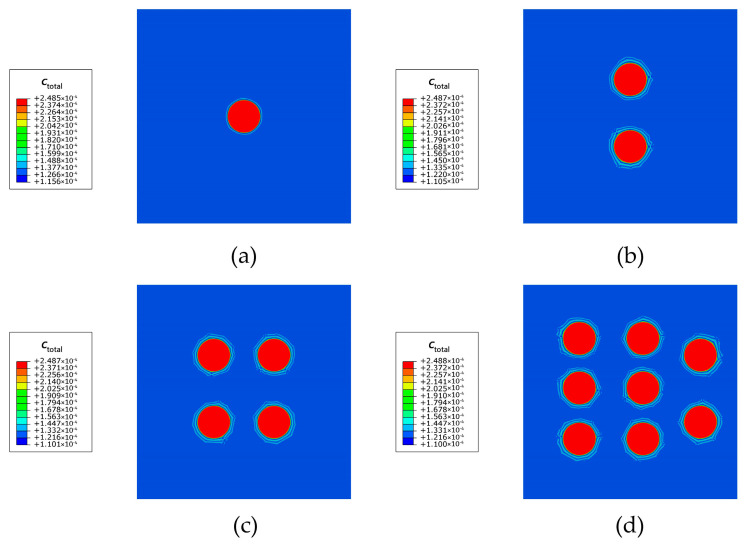
The total hydrogen concentration $c_{\mathrm{total}}$ at $2.17\times 10^{-5}\;Ts$ for Al crystals with (**a**) 1 secondary phase; (**b**) 2 secondary phases; (**c**) 4 secondary phases; and (**d**) 8 secondary phases.

**Figure 15 materials-16-03189-f015:**
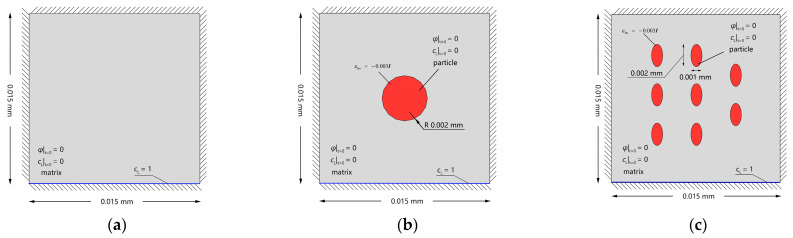
Schematic description of initial and boundary conditions of Al single crystals: (**a**) without secondary phases; (**b**) with a single secondary phase; and (**c**) with refined secondary phases (the total volume fraction equal to that of the single secondary phase).

**Figure 16 materials-16-03189-f016:**
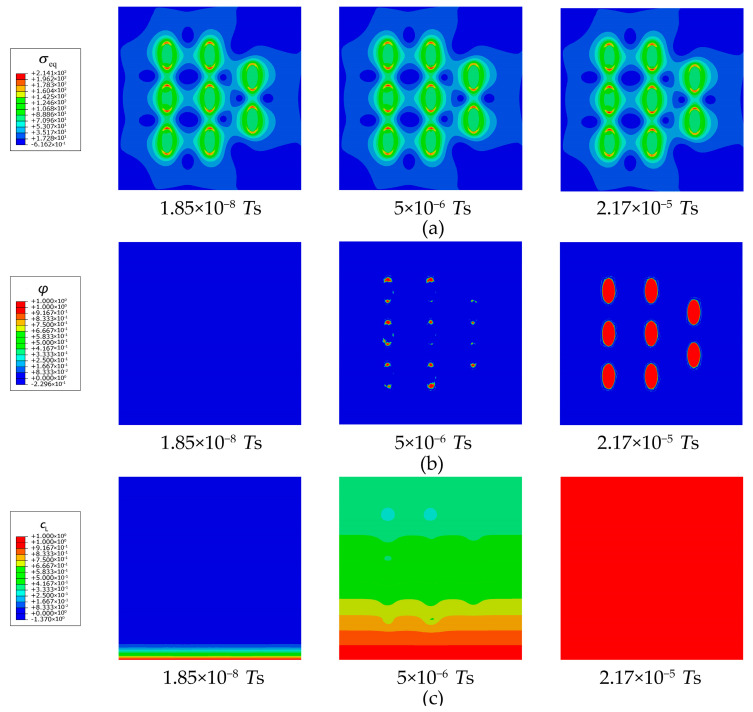
Evolution of (**a**) equivalent stress (unit: MPa), (**b**) occupancy of hydrogen trapping site φ, (**c**) lattice hydrogen concentration $c_{L}$, and (**d**) total hydrogen concentration $c_{\mathrm{total}}$ for the Al crystals with refined secondary phases.

**Figure 17 materials-16-03189-f017:**
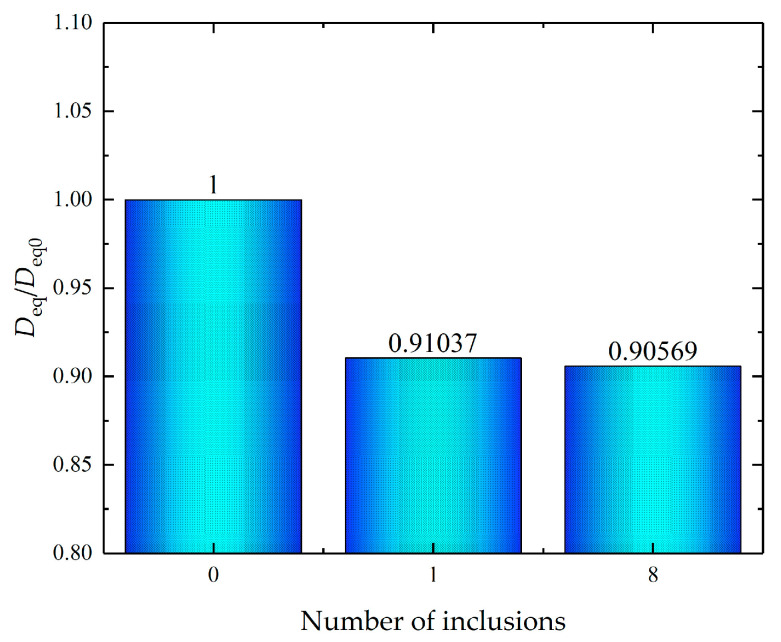
Equivalent diffusive coefficient versus refined number of secondary phases.

**Figure 18 materials-16-03189-f018:**
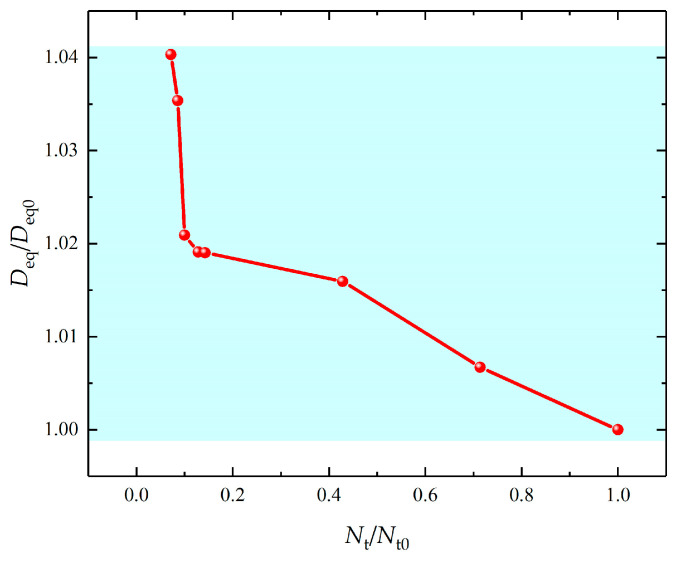
Equivalent diffusive coefficient versus dimensionless number of trapping sites per unit volume $N_{t}/N_{t0}$.

**Figure 19 materials-16-03189-f019:**
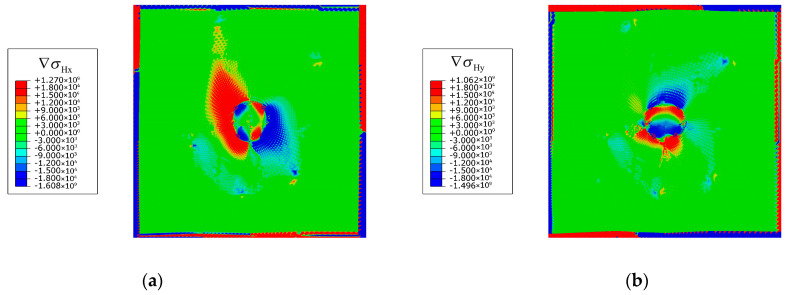
Contours of hydrostatic pressure gradient (unit: MPa/mm) components: (**a**) $\nabla \sigma_{\mathrm{Hx}}$ and (**b**) $\nabla \sigma_{\mathrm{Hy}}$.

**Table 1 materials-16-03189-t001:** Material parameters for hydrogen trapping.

Parameters	Value	Unit
Young’s modulus E	71,000	MPa
Poisson’s ratio μ	0.33	-
Interfacial characteristic thickness $l_{c}$	0.005	mm
Mobility for lattice hydrogen transport D	$2.2\times 10^{-8}$	${\mathrm{mm}}^{2}/s$
Mobility for trap hydrogen transport M	$9\times 10^{-10}$	${\mathrm{mm}}^{2}/\left(N\cdot s\right)$
Free energy density curvature A	53.5	${N/\mathrm{mm}}^{2}$
Surface energy density $G_{c}$	$1.031\times 10^{-3}$	${\mathrm{mJ}/\mathrm{mm}}^{2}$
Number of lattice atoms per unit volume $N_{L}$	$8\times 10^{19}$	$m^{-3}$
Number of lattice atoms per unit volume $N_{T}$	$7\times 10^{19}$	$m^{-3}$
Trapping binding energy $E_{b}$	$11.5\times 10^{6}$	mJ/mol
Saddle point energy $E_{s}$	$2\times 10^{2}$	mJ/mol
The partial molar volume of hydrogen at lattice $V_{\mathrm{hl}}$	$2.1\times 10^{3}$	${\mathrm{mm}}^{3}/\mathrm{mol}$
The partial molar volume of hydrogen at trap $V_{\mathrm{ht}}$	$2.1\times 10^{3}$	${\mathrm{mm}}^{3}/\mathrm{mol}$
Avogadro constant $N_{A}$	$6.02\times 10^{23}$	L/mol
Gas constant R	8314	$\mathrm{mJ}/\left(K\cdot \mathrm{mol}\right)$
Absolute temperature T	300	K
Parameter of diffusive interface $k_{g}$	1	-

## Data Availability

The main data supporting the findings of this study are available within the article. Extra data are available from the corresponding author upon reasonable request.

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
