# Peer review of "Phase-Field Insights into Hydrogen Trapping by Secondary Phases in Alloys"

_materials, 2023, doi:10.3390/ma16083189_

Round 1

Reviewer 1 Report

The Manuscript under review presents an interesting approach to the problem of modeling hydrogen storage materials properties. Although not explicitly stated, the authors have drawn an analogy between the behavior of a material under stress, where cracks and cavities form, and the process of hydrogen loading/release. The hydrogen trapping, enrichment, and storage in the particles of alloys are viewed as a phenomenon coupled with stress concentration due to particular particle geometry. Roughly speaking, instead of cracking and fracture we have a hydrogen accumulation. ABAQUS UMAT user subroutine for implementing a generalized version of phase field fracture allowing for any fracture driving force split (including Drucker-Prager) and any choice of crack density function can be found as open source at https://www.empaneda.com/codes/.

In the manuscript introduction (a list of hydrogen storage technologies) does not contain information about liquid organic hydrogen carriers (LOHCs).  Due to reversibility of hydrogen loading and evolution processes as well as high energy efficiency this direction seems to be quite promising. Data on achievements in this area should be added in the introduction ([10.1016/j.jpowsour.2018.04.011], [10.1039/d2dt00332e]).

It will be very useful to see the approach developed in the Manuscript (the user element subroutine (UEL) of the commercial FE software ABAQUS, p. 5, line 159) also as an open source code and discuss similarities or differences. Some references on the methods applied are required, i.e. https://arxiv.org/abs/2208.10356 etc.

Figure 2 of the manuscript is important, but difficult to read. Authors should either improve the resolution of the figure or split it into two separate ones.

The meaning and origination of the model parameters should be clearly discussed. For example, Table 1. Material parameters for hydrogen trapping - needs to be clarified. It seems that parameters listed there are belongs to aluminum alloys, judging by Young’s modulus E value - 71 MPa (close to well known values). It is not clear where to find binding energy Eb (Figure 4. Free energy curve of hydrogen trapping system) or energy barrier Es.

These values are specific to  aluminum alloys or not? It will be very informative to compare results obtained, for example, for other well-studied systems, i.e. Ni5La. 

These issues are very important. It is hoped that they will be addressed in a revised version of the manuscript.

Reviewer 2 Report

The manuscript involves modeling hydrogen trapping in Al alloys. Though the results are interesting authors should address the following issues.

1 The title looks generic. It needs to be changed.

2. It is not clear which material is being considered for this study. Table 1 gives no clue about the material. Also the data source/reference is also not provided.

3. Further, it is unclear the type of particles used. Where is its position in the lattice is also not mentioned. It is provided that the particles result in the tensile stress field. But, that depends on the size of the particle and its position in the lattice. The details should be provided so to be more meaningful.

4. I think in alloys it is inappropriate to mention them as particles dispersed in the matrix. as alloys are solid solutions and they are not composites. 

5. Whether Eb is the binding energy? The line also seemed to be displaced in the figure. It is not showing the exact binding energy value.

6. Addition of a large number of particles will lead to volumetric strain and structural instability. It is not clear how it is addressed in the present study.

7. The solution to Fick's second law (Eq.8) is based on certain assumptions. Are they valid in this study also? To my understanding, the trend shown in the  diffusion coefficient is the outcome of the diffusion equation that you have chosen. Hence, I'm unable to understand the significance of the result. Fig. 17

8. Please check the figure numbers mentioned in the text. Some of them are incorrect. For example, it is mentioned Fig. 8 corresponds to the tensile stress field. But, it should be fir. 9

9. Units are not provided in any of the figures. Without all the figures doesn't make any sense.

Round 2

Reviewer 1 Report

The manuscript can be accepted for publication in its current form.